# AutoMOAE: Multi-Objective Auto-Algorithm Evolution

## Abstract

Algorithm design has traditionally relied on expert intuition, making it time-consuming and often unable to balance solution quality with computational efficiency. Although LLM-driven methods have shown remarkable progress in automating code synthesis, they seldom address multiple requirements simultaneously. Ignoring such multi-objective trade-offs greatly undermines practical applicability, as real-world algorithm design inevitably involves reconciling competing goals. However, balancing multiple requirements is inherently challenging, often leading to infeasible strategies or unintended side effects during the evolutionary process.

We propose AutoMOAE (Auto-Algorithm Evolution for Multi-Objective requirements), a framework that explicitly incorporates multiple demands into the design process. AutoMOAE leverages LLM prompting to dynamically synthesize crossover and mutation operators augmented with analytical modules, effectively reducing nonproductive optimization steps. Verification operations further ensure strict adherence to both syntactic and functional correctness. Evaluations on graph coloring and Traveling Salesman benchmarks show that AutoMOAE-generated algorithms consistently match or surpass expert-crafted solutions in both solution quality and computational efficiency. These results demonstrate the necessity and promise of integrating multi-objective considerations into automated algorithm design, paving the way for scalable, high-performance synthesis frameworks.

## 1 Introduction

Algorithm design (Kant, 1985) is a cornerstone of computer science, yet traditional approaches heavily rely on expert intuition and suffer from inefficiency and limited generalizability (Ma et al., 2025). The rise of large language models (LLMs) has opened new opportunities for automating this process, making LLM-driven algorithm design an emerging trend (Yang et al., 2024; Liu et al., 2023; Lange et al., 2024).

However, algorithm design is inherently a multi-objective problem, requiring careful trade-offs between solution quality and computational efficiency. Current LLM-driven frameworks largely overlook this characteristic, often focusing on single objectives. As a result, they struggle to balance competing goals, frequently generating infeasible strategies or requiring extensive post hoc corrections, which severely limits their practical applicability.

To address this challenge, we propose AutoMOAE (Auto-Algorithm Evolution for Multi-Objective Requirements), a unified framework that integrates LLM-based code synthesis with evolutionary optimization under explicit multi-objective principles. By dynamically generating crossover and mutation operators tailored to multiple requirements, AutoMOAE not only reduces unproductive search steps but also ensures syntactic and functional correctness through verification.

Evaluations on graph coloring and the traveling salesman problem demonstrate that AutoMOAE consistently matches or surpasses expert-crafted solutions across both solution quality and efficiency, while adapting seamlessly across problem domains. These results underscore the necessity of embedding multi-objective considerations into automated algorithm design and highlight the promise of combining LLM capabilities with evolutionary principles for scalable and high-performance synthesis.

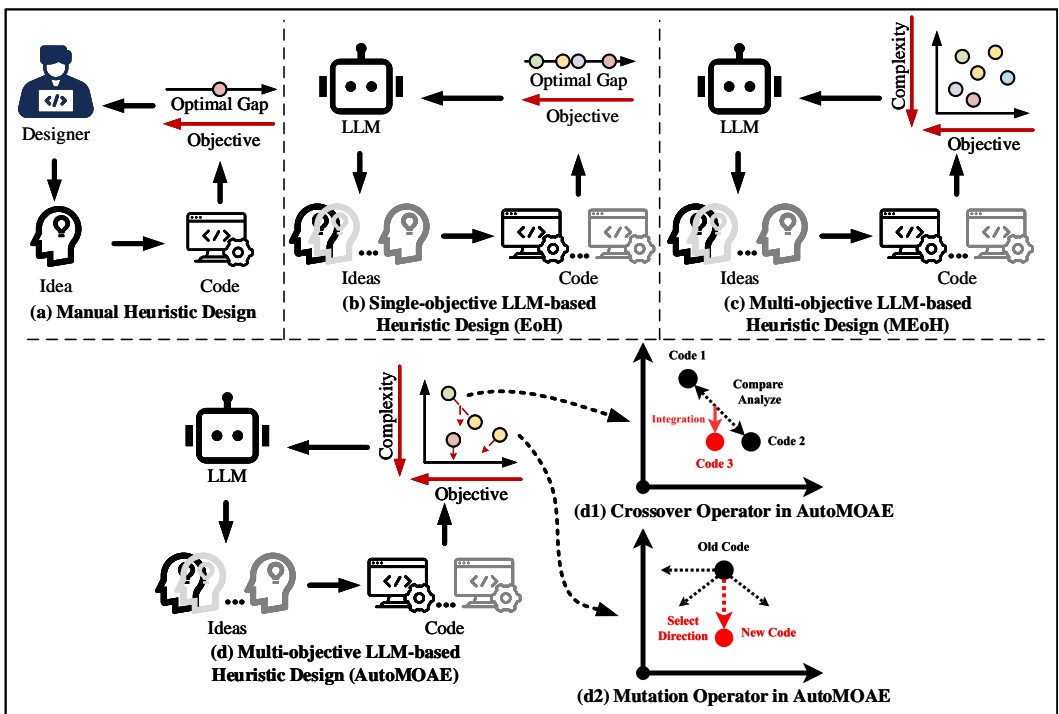

Figure 1: Comparison to human design and existing LLM-based heuristic design. (a) manual heuristic design by human experts, (b) single-objective LLM-based heuristic design (e.g., FunSearch and EoH), (c) multi-objective heuristic design (MEoH), and (d) our multi-objective heuristic design (AutoMOAE), where (d1) Mutation in AutoMOAE. Allowing LLMs to mutate code directly often produces infeasible or harmful changes. AutoMOAE adds an analysis step to constrain mutations toward feasible optimization directions, reducing ineffective variations, (d2) Naïve LLM-based crossover may collapse one code into another. AutoMOAE employs comparative analysis to guide integration, preserving design diversity while ensuring effective crossover.

## 2 RELATED WORKS

### 2.1 AUTOMATED ALGORITHM DESIGN

Prior to the integration of LLMs into algorithm design, extensive research had already explored various approaches to automated algorithm design. These approaches can be broadly categorized into two main directions.

The first direction focuses on optimizing existing heuristic algorithms by searching for optimal parameter configurations to adapt them to specific problems. This line of research has been widely applied to areas such as multi-objective ant colony optimization algorithms (Birattari et al., 2010; Lopez-Ibanez & Stutzle, 2012), stochastic local search methods (Pagnozzi & Stützle, 2019), SAT solvers for flow shop scheduling problems and the traveling salesman problem (TSP) (Hutter et al., 2007; 2009), as well as mixed-integer programming solvers for TSP and vehicle routing problems with time windows (Adamo et al., 2017). While these methods have proven effective in improving performance for specific tasks, their reliance on parameterized frameworks significantly limits their applicability to a single problem type, restricting their generalizability.

The second direction focuses on recombining existing heuristic algorithm components to create new algorithms find a suitable algorithm (Guo et al., 2024; Ma et al., 2024; 2025; Guo et al., 2025). These frameworks conceptualize algorithm design as a combinatorial optimization problem, where modular algorithmic components are assembled to construct optimization algorithms that can adapt to diverse problem settings. Prominent examples include GCOP (Qu et al., 2020), HyFlex (Ochoa et al., 2012), and EvoHyp (Pillay & Beckedahl, 2017), which have garnered significant attention

in the field. Additionally, frameworks like Hydra (Xu et al., 2010) have extended this approach by simultaneously optimizing both algorithm components and parameter configurations, offering a more comprehensive strategy for algorithm design.

## 2.2 LLM DRIVEN ALGORITHM DESIGN

With the advent of LLMs, algorithm design has undergone a transformative change, benefiting from enhanced interpretability and access to an expansive search space. Among the early efforts, the FunSearch (Romera-Paredes et al., 2024) framework pioneered the use of evolutionary computation techniques for code generation, targeting classical mathematical problems. Building on this foundation, researchers introduced a series of frameworks that leverage LLMs to simulate the evolutionary algorithm process for algorithmic code generation. These frameworks include ReEvo (Ye et al., 2024) and EoH (Liu et al., 2024), which focus on single-objective heuristic generation, and MEoH (Yao et al., 2025), which extends this approach to multi-objective heuristics. EoH and MEoH specialize in generating the core components of heuristics. This targeted approach enables them to address more complex tasks, including scientific discovery, combinatorial optimization, and machine learning-related problems. As shown in Figure 1, MEoH utilizes a strength-diversity mechanism for effective population management and selection to balance multiple objectives. In contrast, AutoMOEA not only manages the population at a macro-level using Pareto-based principles, but also embeds analytical modules within its specific crossover and mutation operators. This approach combines macro-level control with local guidance, more effectively directing the population's search toward a balanced solution across multiple objectives.

## 3 METHODS

AutoMOAE represents each candidate algorithm as a code snippet (individual) and iteratively refines a population of such individuals via initialization, genetic variation (crossover and mutation), and Pareto-Based population maintenance.

### 3.1 DIVERSITY-PRESERVING POPULATION INITIALIZATION

To maximize exploratory potential from the outset, AutoMOAE initializes its population in two phases:

**Idea Generation:** Invoke an LLM to produce $N$ succinct algorithmic $idea_i$, each a one-sentence description of a novel design pattern (e.g., greedy nearest-neighbor with adaptive backtracking).

$$idea_i = \textbf{LLM}(P_{think}, N), \ \ i = 1, \cdots, N \tag{1}$$

where $\textbf{LLM}()$ is the large language model interaction function, and the prompt $P_{think}$ instructs the model to perform the following task:

> *Please generate $N$ different algorithmic solution ideas for the following problem: {Problem description}. Requirements: 1) Each idea should be described in one concise line. 2) Do not include any additional information. 3) The ideas should be clearly distinct from each other. 4) Return exactly $N$ lines in total.*

**Code Instantiation:** For each concept, call the LLM with a prompt that expands the description into compilable code. And verify that the code format meets the requirements.

$$\mathcal{X}^0 = \textbf{LLM}(P_{verify}, \textbf{LLM}(P_{run}, \{idea_i\}_{i=1}^N)) \tag{2}$$

where the prompt $P_{run}$ instructs the model to perform the following task:

> *Please provide a Python code implementation for the described problem using the following algorithmic idea: Problem Description: {Problem description}. Idea: $idea_i$. Specific Requirements: 1) Function Name: {function_name}. 2) Input: {input_fmt}. 3) Return Value: {output_fmt}*

The prompt $P_{verify}$ instructs the model to perform the following task:

> *Please check if the following code meets the requirements, and if not, correct it:{code}. Requirements: 1) Only include the function implementation code, without any descriptions, comments, or examples. 2) The main function name must be {function_name}. 3) Do not include a main function or test code. 4) Return only the corrected code, without any other content.*

This two-stage approach ensures high semantic diversity among initial candidates and guards against premature convergence due to superficially similar implementations. These individuals have their fitness evaluated by the evaluator $E(\mathcal{X}^0) = [O_1(x_i^0), ..., O_m(x_i^0)]_{i=1}^N$ ($O_i$ is the objective function) and are then sorted using a non-dominated sorting algorithm, which assigns each a rank, $rank(\mathcal{X}^g) \in 1, \cdots, N$. Subsequently, a selection operator $Selcet()$ randomly selects $k$ individuals $\mathcal{X}_{select}$ from the sub-population with the lowest rank to proceed with the evolutionary process.

## 3.2 LLM-DRIVEN GENETIC OPERATORS

Unlike prior methods that sample from fixed component libraries, AutoMOAE crafts crossover and mutation operators on-the-fly via LLM prompts, tailored to multi-objective algorithm design. Both operators incorporate an automatic validation step, a lightweight static and dynamic checker, that guarantees syntactic correctness and basic functional integrity before evaluation.

### 3.2.1 CROSSOVER OPERATOR

For algorithmic code, the core principle of the crossover operation is to combine the design ideas of individuals $X_{select}$ to generate a new individual. The specific expression is as follows:

$$\mathcal{X}_{off\_crossover} = \mathbf{LLM}(P_{verify}, \mathbf{LLM}(P_{crossover}, Crossover_{Analyse}(\mathcal{X}_{select}))) \qquad (3)$$

**Objective Analysis - $Crossover_{Analyse}()$:** 1) Compute Pareto metrics (e.g., solution quality vs. runtime) for two parent algorithms. 2) Select an objective, such as minimizing runtime, as the 'guide' for crossover. Provide the optimization {target objective} and the {strengths and weaknesses} of the parent generation required for crossover operator.

**Prompt Construction - $P_{crossover}$:** Embed the parents code and the selected design rationale into an LLM prompt that instructs:

> *Please perform a multi-objective optimization crossover based on the following two parent algorithms, with a special focus on the objective {target objective}: Parent 1 : {strengths and weaknesses}. Parent 2 : {strengths and weaknesses}. Please generate a new algorithm with the following requirements: 1) Preserve the advantages of each parent on their respective strength objectives. 2) Specifically optimize performance for the {target objective} objective. 3) Ensure the code is complete and meets the problem's requirements. 4) The function name must be: {function_name}. Return only the final Python code, without any explanation.*

**Synthesis & Validation:** 1) Generate offspring code snippets via the LLM. 2) Since direct code-level crossover can result in syntax errors or functional anomalies, we pass the result through a syntax checker and quick test harness; reject or auto-repair any failures, more details can be see $P_{verify}$.

### 3.2.2 MUTATION OPERATOR

The mutation operator in AutoMOAE, aligned with the crossover operator's design philosophy, is tailored for multi-objective algorithm design by prioritizing the improvement of a single performance metric rather than optimizing all objectives simultaneously.

$$\mathcal{X}_{off\_mutation} = \mathbf{LLM}(P_{verify}, \mathbf{LLM}(P_{mutation}, Mutation_{Analyse}(\mathcal{X}_{select}))) \qquad (4)$$

**Objective Analysis - $Mutation_{Analyse}()$:** Analyze the current individuals Pareto-front position to identify its weakest objective. Return the following information

- Calculate improvement potential: For each optimization target, it calculates the {improvement potential}. Potential is the gap between an individual's current score and the best score ever achieved for that target.

- Identify mutation targets: Locate the target with the highest potential for improvement and designate it as {target_objective}. This instructs the optimization algorithm where its next mutation or variation should be directed to achieve the most significant enhancement.

**Mutation Prompt - $P_{mutation}$:** Craft a prompt directing the LLM to apply a focused modification, including introducing new concepts to solve the problem, adjusting algorithm parameters, or restructuring the code while retaining the original design idea. The details are as follows:

> *Please perform a multi-objective optimization mutation on the following algorithm, focusing on improving the objective {target_objective} (current {improvement potential}. Original Code: {code}. Please: 1) Maintain performance on the other objectives. 2) Specifically optimize the {target_objective} objective. 3) You may introduce new mathematical concepts or optimization methods. 4) Ensure the code is complete and executable. 5) Avoid using recursion or limit its depth. 6) The function name must remain: {function_name}.*

**Synthesis & Validation:** Produce mutated code and validate as in crossover, ensuring functional soundness before acceptance, more details can be see $P_{verify}$.

By embedding validation within each operator, AutoMOAE maintains a pool of executable, diverse algorithms that faithfully explore the multi-objective design space. Each generation of the population can be represented as follows:

$$\mathcal{X}^{g+1} = \textbf{SelectElite}(\mathcal{X}^g \cup \mathcal{X}_{off\_crossover} \cup \mathcal{X}_{off\_mutation}, N) \tag{5}$$

The specific population maintenance process $SelectElite()$ is described in the following section.

### 3.3 PARETO-BASED POPULATION MAINTENANCE

After generating offspring, AutoMOAE evaluates every individual on a standardized benchmark suite, measuring: **Fitness**: quality of solutions (e.g., objective optimality) and **Runtime**: wall-clock execution time. The fitness value reflects the quality of the solution, such as the optimality of the objective function, while runtime measures the computational efficiency of the algorithm.

#### 3.3.1 PARETO FRONT CONSTRUCTION

The selection process in AutoMOAE is guided by the Pareto front, which identifies individuals that achieve an optimal trade-off between fitness and runtimei.e., those not dominated by any others across all objectives. By prioritizing these individuals for retention, the framework promotes convergence toward globally optimal solutions while preserving objective balance. This strategy maintains population diversity and mitigates premature convergence to local optima.

**First Front**: identify all non-dominated individuals (no other candidate is strictly better in both fitness and runtime).

**Subsequent Fronts**: iteratively extract the next layer of non-dominated individuals from the remaining pool until the desired population size is reached.

#### 3.3.2 REPLACEMENT STRATEGY

For newly generated individuals, their fitness values and runtimes are compared against those of the individuals on the current Pareto front.

**Dominance Replacement**: New offspring are first compared to the current Pareto front: if an offspring dominates a front member (i.e., is no worse in all objectives and strictly better in at least one), it replaces that member; if it is dominated by every front member, it is discarded.

---

**Algorithm 1** AutoMOAE

---

**Require:** Population size $N$, Population generations $G$
**Ensure:** The final optimized answer $y$
 1: // Initialization
 2: $\{idea_i\}_{i=1}^N \leftarrow \mathbf{LLM}(P_{think})$.
 3: $X^0 \leftarrow \mathbf{LLM}(P_{verify}, \mathbf{LLM}(P_{run}, \{idea_i\}_{i=1}^N))$.
 4: $E(\mathcal{X}^0) \leftarrow [O_1(x_i^0), ..., O_m(x_i^0)]_{i=1}^N$ // Individual Fitness Assessment. $O_i()$
 5: $\mathcal{F}^0 \leftarrow \{E(\mathcal{X}^0), \mathcal{X}^0\}$ // Constructing the Pareto Frontier, $E()$ is the evaluation function
 6: **for** $g = 0$ to $G - 1$ **do**
 7: $\quad \mathcal{X}_{select} = \mathbf{Select}(\mathcal{X}^g, k)$ // $k$ is the number of individuals selected.
 8: $\quad \mathcal{X}_{off\_crossover} = \mathbf{LLM}(P_{verify}, \mathbf{LLM}(P_{crossover}, Crossover_{Analyse}(\mathcal{X}_{select})))$
 9: $\quad \mathcal{X}_{off\_mutation} = \mathbf{LLM}(P_{verify}, \mathbf{LLM}(P_{mutation}, Mutation_{Analyse}(\mathcal{X}_{select})))$
10: $\quad$ // Update candidates
$\quad\quad \mathcal{X}^{g+1} = \mathbf{SelectElite}(\mathcal{X}^g \cup \mathcal{X}_{off\_crossover} \cup \mathcal{X}_{off\_mutation}, N)$
11: $\quad$ // Constructing the Pareto Frontier,
$\quad\quad \mathcal{F}^{g+1} = \big\{ x \in X^{g+1} \mid x \text{ is non-dominated in } \mathcal{X}^{g+1} \big\}$
12: **end for**
13: $y_{final} \leftarrow$ Select the highest-scoring response from $F_1$.
14: **Return** $y$

---

**Size Enforcement**: Should the population fall below the target size, additional Pareto fronts are constructed hierarchically from the remaining candidates until the size threshold is met or the candidate pool is exhausted.

This layered selection preserves diversity while prioritizing high-quality solutions, broadening the search space and improving the likelihood of global optimal convergence. Ultimately, the Pareto frontier $\mathcal{F} = \{\mathcal{F}_1, \cdots, \mathcal{F}_m\}$ can be expressed as follows:

$$\mathcal{F}^{g+1} = \big\{ x \in \mathcal{X}^{g+1} \mid x \text{ is non-dominated in } \mathcal{X}^{g+1} \big\}, \tag{6}$$

The complete steps of AutoMOAE are detailed in Algorithm 1.

## 4 EXPERIMENTS

### 4.1 EXPERIMENTAL SETTINGS

**Problems** We selected several classical problems to evaluate the effectiveness of AutoMOAE in algorithm design, including the Traveling Salesman Problem (TSP) (Lin, 1965) and the Graph Coloring Problem (GCP) (Matula et al., 1972). The TSP is a combinatorial optimization problem that seeks the shortest possible route visiting a set of cities exactly once and returning to the starting point. The GCP involves assigning colors to the vertices of a graph such that no two adjacent vertices share the same color, while minimizing the total number of colors used.

**Datasets and Implementation Details** To ensure a fair comparison of algorithms developed by different automated algorithm design frameworks, we used randomly generated problem instances to assist in testing the algorithms produced by each framework. For both problems, the best algorithm designed by each framework was selected for subsequent performance evaluation. For the TSP, a manually constructed problem instance with 100 nodes was used as the training data across all frameworks during the algorithm development process, while unweighted path instances from the TSPLIB dataset (Reinelt, 1991) were selected as testing data. For the GCP, a set of 12 problem instances was manually created, with each instance name consisting of three components: the problem name, the number of nodes, and the graph density metric. The dataset was generated using classical random graph generation models, adjusting the number of vertices and edges to produce graphs with varying densities and average degrees. Additional processing details are provided in **Appendix A.2**.

During the algorithm design process, a single problem instance with 125 nodes and a density of approximately 0.5 was fixed as the training data, while all 12 instances were used for testing. For all frameworks, GPT-4o-mini was used as the base model, with a population size of 8 and an

evolution limit of 10 generations. For multi-objective frameworks such as AutoMOAE and MEoH, the algorithm selected for performance comparison was the one achieving the best fitness metric in the final population, without prioritizing runtime as the primary criterion.

**Baseline**   For GCP, the baseline algorithms include the Greedy algorithm, the Welsh-Powell algorithm (Olariu & Randall, 1989), and the DSATUR algorithm (San Segundo, 2012). For TSP, we used the Greedy algorithm, the nearest neighbor algorithm (NN), and the insertion method. Additionally, we included automated algorithm design frameworks such as FunSearch, EoH, and MEoH in the comparison.

## 4.2 Algorithm performance

**GCP**   For GCP, we compared multiple classical algorithms designed by human experts with the best algorithms developed by different frameworks. The algorithm developed by AutoMOAE achieved excellent performance in both solution quality and runtime. The metric used to evaluate algorithm performance was the number of colors required to completely color a graph with a given number of nodes. The results are presented in Figure 2a, Tables 4 and 5 of **Appendix A.3**. From Figure 2a, it can be observed that the algorithm developed by AutoMOAE allocated colors using significantly less time compared to EoH and DSATUR, while achieving the same number of colors. Analysis of the algorithm code of **Appendix A.5** revealed that both AutoMOAE and EoH ultimately designed algorithms based on the DSATUR framework. However, the key difference lies in their optimization strategies: AutoMOAE shifted its focus to reducing computational overhead after identifying diminishing returns in further minimizing the number of colors, whereas EoH continued to attempt integrating new components to improve solution quality, which resulted in longer runtime compared to DSATUR.

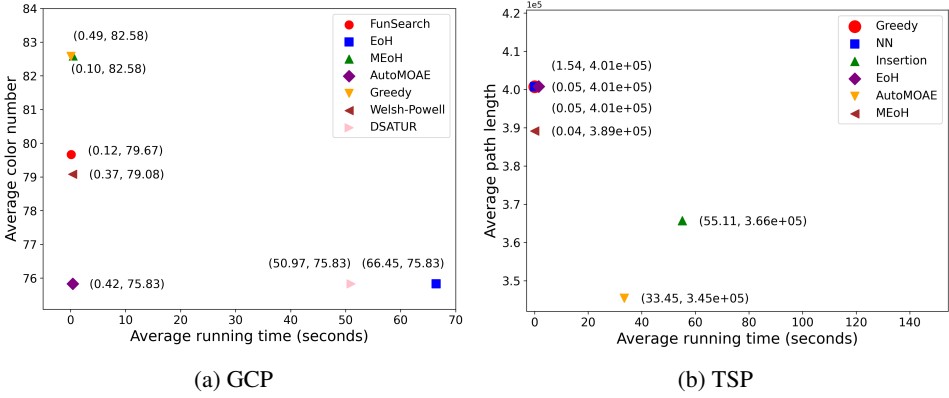

(a) GCP                    (b) TSP

Figure 2: Average performance metrics of different algorithms on GCP and TSP instances.

**TSP**   For TSP, due to the large number of instances and significant variation in performance metrics across different problem instances, we used ranking-based metrics to evaluate algorithm performance. Specifically, the rank achieved by each algorithm on individual instances was averaged across all instances in the TSPLIB dataset, and the results are summarized in Table 1 and Figure 2b. The results show that the algorithm developed by AutoMOAE achieved the shortest path length in the majority of instances, with an average rank of 1.26, significantly outperforming other automated algorithm design frameworks. However, the runtime performance of the AutoMOAE-developed algorithm was less favorable. This is primarily due to the design of AutoMOAEs crossover and mutation operators, which can introduce new concepts or techniques into the algorithm. The detailed results for AutoMOEA and the various baselines on each specific instance of the TSPLIB, including solution quality and runtime, are provided in Tables 6 and 7 of **Appendix A.4**. The specific implementation details of the algorithm are provided in Appendix A.5.

Table 1: Average rank of different algorithms on performance metrics in TSPLIB.

|  | Greedy | NN | Insertion | EoH | MEoH | AutoMOAE |
|---|---|---|---|---|---|---|
| Path Length (Rank) | 3.77 | 3.77 | 2.73 | 4.04 | 3.24 | **1.26** |
| Runtime (Rank) | 1.61 | **1.50** | 5.31 | 4.09 | 2.92 | 5.39 |

Table 2: The proportion of new-generation individuals within the elite population.

|  | Gen1 | Gen2 | Gen3 | Gen4 | Gen5 | Gen6 | Gen7 | Gen8 | Gen9 | Gen10 | Avg. |
|---|---|---|---|---|---|---|---|---|---|---|---|
| w/ Analysis Component | 25.0% | 12.5% | 32.5% | 62.5% | 62.5% | 75.0% | 32.5% | 50% | 50% | 0.0% | 40.25% |
| w/o Analysis Component | 32.5% | 32.5% | 75.0% | 12.5% | 0.0% | 0.0% | 0.0% | 0.0% | 0.0% | 0.0% | 15.25% |

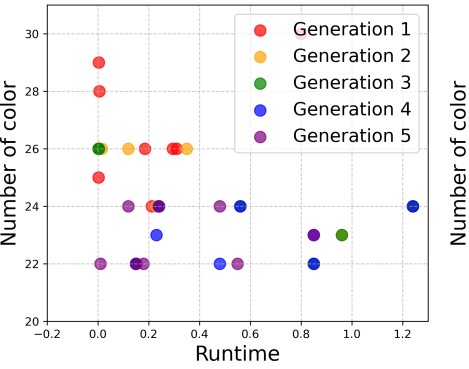 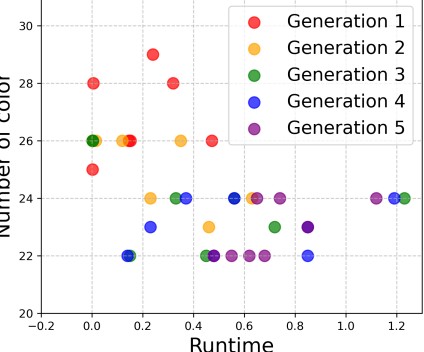

(a) w/ Analysis Component         (b) w/o Analysis Component

Figure 3: The impact of analysis component in genetic operators on population distribution. (a) w/ Analysis Component; (b) w/o Analysis Component.

## 4.3 ABLATION STUDY

To demonstrate the effectiveness of the genetic operators designed in AutoMOAE, we conducted an ablation study on the analysis component within these operators. Specifically, we compared genetic operators with and without the analysis component in the algorithm population's evolutionary process. The distribution of newly generated algorithm populations (prior to population selection) in the objective space was recorded and visualized in Figure 3. For clarity in observing the differences in individual distributions across generations, only the first five generations were selected for visualization. By comparing the code characteristics and performance distributions of each generation, we observed that with the analysis component, most individuals effectively shifted their optimization focus to algorithm runtime when further improvement in the number of colors was no longer feasible. In contrast, without the analysis component, individuals in the population lacked a clear optimization direction and continued attempting to reduce the number of colors by introducing additional strategies. This resulted in poor runtime performance and, in many cases, regression in algorithm performance compared to the previous generation.

Additionally, as shown in Table 2, we tracked the proportion of individuals from each new generation that successfully entered the elite population. It is evident that when the analysis component was retained, this proportion was significantly higher, indicating that the newly generated individuals were more competitive. In contrast, without the analysis component, the average proportion dropped from 40.25% to 15.25%, and the evolutionary process even stagnated in its later stages.

## 4.4 ALGORITHM COMPLEXITY

In frameworks utilizing LLMs for algorithm design, many approaches, such as EoH and MEoH, simplify the inherently complex task of algorithm design by focusing solely on the development of core components within heuristic algorithms. While this strategy enables these frameworks to tackle more complex problems efficiently, it also constrains them to the heuristic algorithm paradigm,

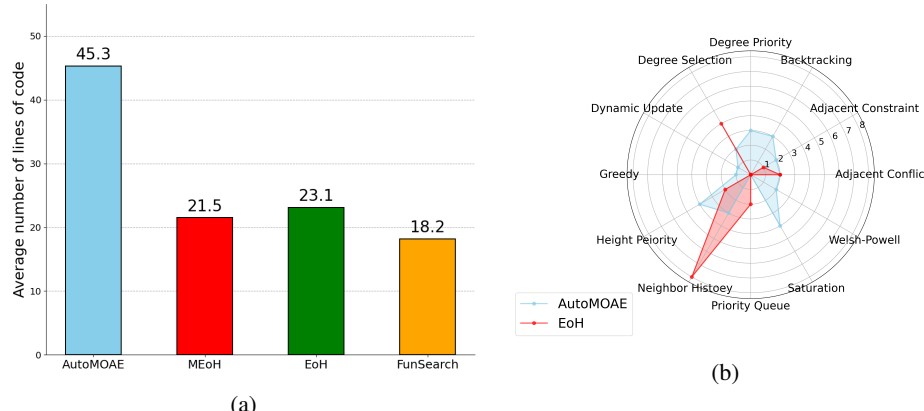

(a)                                           (b)

Figure 4: (a) The average number of uncommented lines of code in the final algorithm populations, serving as a proxy for algorithmic complexity. (b) The frequency of algorithmic components within the final populations from AutoMOAE and MEoH on the GCP, used as a measure of design diversity.

limiting their ability to explore beyond this framework. To enhance the interpretability of the algorithm design process and increase the capability to address more complex problems, automated algorithm design frameworks should aim to generate complete and sophisticated algorithmic code.

In this evaluation, we used the number of uncommented lines of code as a metric to assess the complexity of the algorithms generated by different frameworks. The results are visualized in the Figure 4a. Comparisons reveal that the average number of code lines in the final algorithm population generated by AutoMOAE is 45.3, significantly higher than that of other automated algorithm design frameworks. This is attributed to AutoMOAEs crossover and mutation operators, which effectively introduce new concepts and techniques into the algorithms. These operations substantially expand the search space for algorithmic solutions and indicate AutoMOAEs considerable potential for solving even more complex problems.

### 4.5 COMPARISON OF DIVERSITY

During the algorithm evolution process, the diversity of algorithm design ideas significantly impacts the evolutionary potential of the population. To evaluate the diversity within algorithm populations, we utilized an LLM to summarize the primary implementation ideas of the eight code individuals from each population into a set of keywords. After removing common high-frequency terms such as "DSATUR," we performed keyword frequency analysis and visualized the results in Figure 4b. According to the keyword frequency analysis, nearly all of the algorithms developed by EoH relied on greedy strategies, with minimal application of other strategies. In contrast, the analysis of AutoMOAE's final population revealed a total of 27 distinct keywords, compared to 19 keywords in MEoH's final population. This result demonstrates that the operators designed in AutoMOAE effectively introduce new strategies, concepts, and components into the population, thereby enhancing its diversity.

### 5 CONCLUSION

This paper introduced AutoMOAE, a novel framework for multi-objective automated algorithm evolution. The core innovation of AutoMOAE is the integration of analytical operators within its crossover and mutation mechanisms. This design mitigates ineffective evolutionary steps by intelligently guiding the optimization process, which significantly enhances the framework's robustness when navigating complex trade-offs. To support its multi-objective capabilities, AutoMOAE employs a two-stage initialization method and a dominance-based selection strategy to preserve population diversity. Our findings establish AutoMOAE as a powerful tool for researchers, enabling the efficient exploration of diverse design concepts and the generation of highly interpretable baseline algorithms. For a more detailed discussion, see **Appendix A.1**.

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

# A  APPENDIX

## A.1  DISCUSSION

**Comparison Between AutoMOAE and Deep Thinking Large Models**  Deep thinking large models, such as DeepSeek-R1 (DeepSeek-AI et al., 2025) or OpenAI o3, have achieved significant success across various domains, including mathematical reasoning and other complex tasks. These models excel by engaging in extensive reflection and iterative exploration prior to executing specific tasks, enabling them to determine effective strategies. As a result, they can partially substitute for certain functionalities of frameworks like AutoMOAE, such as integrating crossover or mutation operators. However, compared to automated algorithm design frameworks, deep thinking models lack the capability to accurately assess whether the generated code or algorithm can solve the target problem, as well as the extent and efficiency of the solution. We propose that deep thinking models can serve as initialization operators for algorithm populations, providing high-quality algorithmic candidates to enhance the diversity and performance of the population in automated design frameworks.

**Differences Between AutoMOAE and Existing Frameworks**  Researchers have proposed several frameworks for automated algorithm design, including FunSearch, EoH, and MEoH. Compared to these frameworks, AutoMOAE introduces distinct motivations and contributions. While all these frameworks, including AutoMOAE, adopt evolutionary algorithm-based approaches for algorithm design, AutoMOAE draws its inspiration from simulating the workflow of human researchers during algorithm development. This approach aims to maximize the potential of LLMs by leveraging their sub-human-level intelligence in a structured and creative manner. In comparison to MEoH, which is also a multi-objective algorithm design framework, AutoMOAE differs significantly in its methodology. AutoMOAE does not impose strict constraints on metrics such as population crowding during evolution. Instead, its multi-objective nature and operational operators are designed to identify components or strategies that can enhance algorithm performance. Furthermore, when one objective becomes difficult to optimize, AutoMOAE shifts its focus to another objective, thereby reducing ineffective iterations during the evolutionary process and improving the efficiency of algorithm evolution.

**Limitations of AutoMOAE**  Despite its strengths, AutoMOAE has certain limitations. One key challenge is handling complex input and output conditions, as LLMs often struggle to accurately interpret parameter sequences and output orders, potentially leading to errors in functionality. To address this, researchers must simplify input-output structures and minimize the number of parameters required for problem instances. Additionally, while AutoMOAEs multi-objective design reduces ineffective iterations, it does not explicitly prioritize maintaining diversity within the algorithm population, which may restrict its ability to explore a broader solution space in some scenarios.

## A.2  DETAILS ON GENERATING DATASETS FOR IMAGE COLORING PROBLEMS

The Graph Coloring Problem (GCP) is a classic problem in graph theory, where the objective is to assign a color to each vertex of a graph such that no two adjacent vertices share the same color, while minimizing the total number of colors used. To validate the effectiveness of the proposed method on the GCP, this study employs a set of artificially generated datasets, designated as the GCP dataset. Each problem instance within this dataset is named using a combination of the dataset identifier, the number of nodes, and a graph density metric. The generation of this dataset is based on classic random graph models. By adjusting the number of vertices and edges, we produced graph instances with varying densities and average degrees to comprehensively evaluate the algorithm's adaptability to both sparse and dense graphs. The fundamental characteristics of these instances are summarized in Table 3.

During the dataset generation process, we first established the vertex counts for graphs of different scales, namely 125, 250, 500, and 1000 nodes. Subsequently, a random graph generation algorithm was utilized to construct graph structures that conform to target densities. To ensure the diversity and representativeness of the generated graphs, three instances with distinct densities were created for each vertex count level, corresponding to sparse (density $\approx 0.1$), medium-density (density $\approx 0.5$), and dense (density $\approx 0.9$) graphs. Furthermore, to simulate the complexity found in real-world

problems, random perturbations were introduced to the generated instances to increase their structural irregularity.

These datasets will serve as the benchmark for evaluating the evolutionary performance of our proposed method on the Graph Coloring Problem. Specifically, one instance was randomly selected to assess algorithmic performance during the evolutionary process itself. The development of the algorithm targets two objectives: minimizing the number of colors used and minimizing the required computation time. After the optimal algorithm has been identified, it will be comprehensively tested on the entire suite of datasets.

Table 3: GCP Instance Basic Information Statistics

| Instance Name | Vertices | Edges | Density | Avg. Degree |
|---|---|---|---|---|
| GCP125.1 | 125 | 736 | 0.0949 | 11.776 |
| GCP125.5 | 125 | 3891 | 0.5021 | 62.256 |
| GCP125.9 | 125 | 6961 | 0.8982 | 111.376 |
| GCP250.1 | 250 | 3218 | 0.1034 | 25.744 |
| GCP250.5 | 250 | 15668 | 0.5034 | 125.344 |
| GCP250.9 | 250 | 27897 | 0.8963 | 223.176 |
| GCP500.1 | 500 | 12458 | 0.0999 | 49.832 |
| GCP500.5 | 500 | 62624 | 0.5020 | 250.496 |
| GCP500.9 | 500 | 112437 | 0.9013 | 449.748 |
| GCP1000.1 | 1000 | 49629 | 0.0994 | 99.258 |
| GCP1000.5 | 1000 | 249826 | 0.5002 | 499.652 |
| GCP1000.9 | 1000 | 449449 | 0.8998 | 898.898 |

## A.3 DETAILED RESULTS OF AUTOMOAE AND BASELINES ON THE GRAPH COLORING PROBLEM (GCP)

As shown in Tables 4 and 5, AutoMOAE demonstrates the best overall performance on the GCP. Its solution quality is comparable to that of DSATUR and EoH, while its runtime is notably faster.

Table 4: Number of colors used by algorithms on different GCP instances.

|            | Greedy | Welsh-Powell | DSATUR | FunSearch | EoH    | MEoH   | AutoMOAE |
| ---------- | ------ | ------------ | ------ | --------- | ------ | ------ | -------- |
| GCP0125.1  | 8      | 7            | **6**  | 8         | **6**  | 8      | **6**    |
| GCP0125.5  | 26     | 23           | **22** | 25        | **22** | 26     | **22**   |
| GCP0125.9  | 56     | 53           | **51** | 55        | **51** | 56     | **51**   |
| GCP0250.1  | 13     | 11           | **10** | 11        | **10** | 13     | **10**   |
| GCP0250.5  | 43     | 41           | **37** | 40        | **37** | 43     | **37**   |
| GCP0250.9  | 99     | 93           | **92** | **92**    | **92** | 99     | **92**   |
| GCP0500.1  | 20     | 18           | **16** | 18        | **16** | 20     | **16**   |
| GCP0500.5  | 72     | 71           | **65** | 71        | 65     | 72     | **65**   |
| GCP0500.9  | 175    | **169**      | 170    | 171       | 170    | 175    | 170      |
| GCP1000.1  | 31     | 29           | **27** | 29        | **27** | 31     | **27**   |
| GCP1000.5  | 127    | 121          | **115**| 124       | **115**| 127    | **115**  |
| GCP1000.9  | 321    | 313          | **299**| 312       | **299**| 321    | **299**  |
| Average Rank | # 5.92 | # 4.00     | **# 1.08** | # 4.17 | **# 1.08** | # 5.92 | **# 1.08** |

Table 5: Algorithm execution time across different GCP instances. Unit: s.

|            | Greedy | Welsh-Powell | DSATUR   | FunSearch | EoH      | MEoH   | AutoMOAE |
| ---------- | ------ | ------------ | -------- | --------- | -------- | ------ | -------- |
| GCP0125.1  | 0.0037 | 0.0037       | 0.3195   | 0.0051    | 0.4083   | 0.0013 | 0.0084   |
| GCP0125.5  | 0.0045 | 0.0051       | 0.3571   | 0.0053    | 0.4813   | 0.0062 | 0.0165   |
| GCP0125.9  | 0.0049 | 0.0113       | 0.3737   | 0.0059    | 0.5863   | 0.0164 | 0.0256   |
| GCP0250.1  | 0.0151 | 0.0145       | 2.5754   | 0.0191    | 3.0949   | 0.0048 | 0.0275   |
| GCP0250.5  | 0.0166 | 0.0276       | 2.7750   | 0.0212    | 3.7176   | 0.0262 | 0.0638   |
| GCP0250.9  | 0.0194 | 0.0584       | 2.9210   | 0.0222    | 4.2889   | 0.1001 | 0.1037   |
| GCP0500.1  | 0.0656 | 0.0723       | 20.3544  | 0.0825    | 24.5486  | 0.0190 | 0.1113   |
| GCP0500.5  | 0.0699 | 0.1442       | 22.3491  | 0.0855    | 29.4261  | 0.1448 | 0.2955   |
| GCP0500.9  | 0.0803 | 0.3501       | 24.8234  | 0.0913    | 34.0722  | 0.5569 | 0.4656   |
| GCP1000.1  | 0.2631 | 0.3304       | 162.8072 | 0.3162    | 193.7323 | 0.0771 | 0.4666   |
| GCP1000.5  | 0.2955 | 0.8698       | 179.6298 | 0.3590    | 236.0436 | 0.9171 | 1.2506   |
| GCP1000.9  | 0.3155 | 2.5864       | 192.3301 | 0.4782    | 266.9487 | 3.9840 | 2.1976   |
| Average Rank | # 1.42 | # 3.00     | # 6.00   | # 2.67    | # 7.00   | # 3.08 | # 4.75   |

## A.4 DETAILED RESULTS OF AUTOMOAE AND BASELINES ON TSPLIB.

Figures 6 and 7 present the detailed results and computational times for each method on the TSPLIB dataset.

Table 6: Resulting path length and rank (#) for each algorithm across the TSPLIB instances.

| | Greedy | NN | Insertion | EoH-TSP | MEoH-TSP | AutoMOAE-TSP |
|---|---|---|---|---|---|---|
| burma14 | 38.69( # 3) | 38.69( # 3) | 32.44( # 2) | 38.69( # 3) | 38.80( # 6) | 31.21( # 1) |
| ulysses16 | 104.73( # 5) | 104.73( # 5) | 79.39( # 2) | 86.60( # 3) | 104.65( # 4) | 74.00( # 1) |
| ulysses22 | 89.64( # 3) | 89.64( # 3) | 76.99( # 1) | 89.64( # 3) | 91.92( # 6) | 85.64( # 2) |
| att48 | 40526.42( # 5) | 40526.42( # 5) | 37314.09( # 2) | 40364.11( # 4) | 37686.87( # 3) | 34902.00( # 1) |
| eil51 | 513.61( # 4) | 513.61( # 4) | 496.25( # 3) | 513.61( # 4) | 458.95( # 1) | 465.91( # 2) |
| berlin52 | 8980.92( # 3) | 8980.92( # 3) | 9014.89( # 6) | 8980.92( # 3) | 8835.06( # 2) | 8217.14( # 1) |
| st70 | 805.53( # 3) | 805.53( # 3) | 778.99( # 2) | 805.53( # 3) | 871.65( # 6) | 753.15( # 1) |
| pr76 | 153461.92( # 5) | 153461.92( # 5) | 125936.21( # 3) | 145069.74( # 4) | 123787.14( # 2) | 111856.22( # 1) |
| eil76 | 711.99( # 5) | 711.99( # 5) | 612.39( # 2) | 669.24( # 4) | 577.27( # 1) | 622.71( # 3) |
| gr96 | 707.09( # 5) | 707.09( # 5) | 651.44( # 3) | 673.92( # 4) | 573.48( # 1) | 623.53( # 2) |
| rat99 | 1564.72( # 4) | 1564.72( # 4) | 1482.02( # 2) | 1564.72( # 4) | 1492.74( # 3) | 1377.07( # 1) |
| kroC100 | 26327.36( # 4) | 26327.36( # 4) | 25262.17( # 3) | 26327.36( # 4) | 24294.06( # 2) | 23392.80( # 1) |
| kroE100 | 27587.19( # 4) | 27587.19( # 4) | 25900.02( # 3) | 27587.19( # 4) | 25221.45( # 2) | 24282.64( # 1) |
| rd100 | 9941.16( # 3) | 9941.16( # 3) | 8979.37( # 2) | 9941.16( # 3) | 10510.18( # 6) | 8864.57( # 1) |
| kroA100 | 26856.39( # 4) | 26856.39( # 4) | 24307.78( # 3) | 26856.39( # 4) | 22683.29( # 1) | 22830.62( # 2) |
| kroB100 | 29155.04( # 4) | 29155.04( # 4) | 25580.92( # 2) | 29155.04( # 4) | 25679.71( # 3) | 25392.46( # 1) |
| kroD100 | 26950.46( # 4) | 26950.46( # 4) | 25204.27( # 2) | 26950.46( # 4) | 26072.81( # 3) | 24720.72( # 1) |
| eil101 | 825.24( # 4) | 825.24( # 4) | 702.96( # 2) | 844.91( # 6) | 720.41( # 3) | 702.70( # 1) |
| lin105 | 20362.76( # 4) | 20362.76( # 4) | 16934.62( # 1) | 20362.76( # 4) | 19041.58( # 3) | 17762.08( # 2) |
| pr107 | 46678.15( # 2) | 46678.15( # 2) | 52587.76( # 6) | 47029.63( # 4) | 50560.61( # 5) | 45574.46( # 1) |
| pr124 | 69299.43( # 4) | 69299.43( # 4) | 65318.19( # 2) | 69299.43( # 4) | 68371.30( # 3) | 61910.46( # 1) |
| bier127 | 135751.78( # 2) | 135751.78( # 2) | 140690.94( # 6) | 135751.78( # 2) | 138054.23( # 5) | 122109.70( # 1) |
| ch130 | 7575.29( # 4) | 7575.29( # 4) | 7279.21( # 3) | 7575.29( # 4) | 7091.76( # 1) | 7093.18( # 2) |
| pr136 | 120777.86( # 4) | 120777.86( # 4) | 109587.25( # 1) | 118776.81( # 3) | 123247.21( # 6) | 110347.07( # 2) |
| gr137 | 1022.22( # 4) | 1022.22( # 4) | 821.29( # 1) | 1022.22( # 4) | 901.99( # 3) | 849.98( # 2) |
| pr144 | 61650.72( # 3) | 61650.72( # 3) | 73033.13( # 6) | 61650.72( # 3) | 60133.16( # 1) | 61399.21( # 2) |
| kroB150 | 32825.75( # 4) | 32825.75( # 4) | 31588.68( # 3) | 32825.75( # 4) | 29848.14( # 2) | 27674.75( # 1) |
| ch150 | 8194.61( # 4) | 8194.61( # 4) | 7994.34( # 3) | 8194.61( # 4) | 7589.72( # 2) | 7161.32( # 1) |
| kroA150 | 33609.87( # 4) | 33609.87( # 4) | 29966.54( # 2) | 33609.87( # 4) | 31060.89( # 3) | 28613.32( # 1) |
| pr152 | 85702.95( # 2) | 85702.95( # 2) | 88530.82( # 5) | 85702.95( # 2) | 92682.12( # 6) | 80000.73( # 1) |
| u159 | 54669.03( # 4) | 54669.03( # 4) | 49981.41( # 1) | 57436.69( # 6) | 51577.24( # 3) | 50140.36( # 2) |
| rat195 | 2761.96( # 3) | 2761.96( # 3) | 2814.57( # 6) | 2761.96( # 3) | 2575.61( # 2) | 2490.60( # 1) |
| d198 | 18620.07( # 3) | 18620.07( # 3) | 17631.80( # 2) | 18620.07( # 3) | 19454.99( # 6) | 17340.94( # 1) |
| kroA200 | 35798.41( # 5) | 35798.41( # 5) | 35337.51( # 4) | 33901.53( # 3) | 33224.38( # 2) | 31512.83( # 1) |
| kroB200 | 36981.59( # 4) | 36981.59( # 4) | 35421.70( # 3) | 36981.59( # 4) | 34079.08( # 1) | 34372.39( # 2) |
| gr202 | 619.40( # 4) | 619.40( # 4) | 570.14( # 3) | 619.40( # 4) | 559.57( # 2) | 529.30( # 1) |
| ts225 | 152493.55( # 4) | 152493.55( # 4) | 160009.16( # 6) | 146183.10( # 3) | 131454.99( # 1) | 139697.02( # 2) |
| tsp225 | 4829.00( # 5) | 4829.00( # 5) | 4468.20( # 3) | 4786.42( # 4) | 4430.19( # 2) | 4169.09( # 1) |
| pr226 | 94685.45( # 4) | 94685.45( # 4) | 91046.65( # 2) | 94402.09( # 3) | 96212.20( # 6) | 87543.04( # 1) |
| gr229 | 2014.71( # 4) | 2014.71( # 4) | 1825.83( # 2) | 2014.71( # 4) | 1987.56( # 3) | 1764.18( # 1) |
| gil262 | 3241.47( # 4) | 3241.47( # 4) | 2804.23( # 3) | 3259.42( # 6) | 2748.90( # 1) | 2757.03( # 2) |
| pr264 | 58022.86( # 2) | 58022.86( # 2) | 58225.34( # 4) | 58328.28( # 5) | 59000.73( # 6) | 56762.06( # 1) |
| a280 | 3148.11( # 4) | 3148.11( # 4) | 3101.79( # 2) | 3182.09( # 6) | 3123.70( # 3) | 2828.71( # 1) |
| pr299 | 59899.01( # 3) | 59899.01( # 3) | 58124.45( # 2) | 60220.49( # 5) | 61338.05( # 6) | 52408.65( # 1) |
| lin318 | 54033.58( # 4) | 54033.58( # 4) | 49454.81( # 2) | 54033.58( # 4) | 50085.92( # 3) | 49153.11( # 1) |
| linhp318 | 54033.58( # 4) | 54033.58( # 4) | 49454.81( # 2) | 54033.58( # 4) | 50085.92( # 3) | 49153.11( # 1) |
| rd400 | 19168.05( # 4) | 19168.05( # 4) | 18629.98( # 3) | 19168.05( # 4) | 17599.52( # 2) | 16651.71( # 1) |
| fl417 | 15114.12( # 4) | 15114.12( # 4) | 14179.84( # 3) | 15256.42( # 6) | 13680.66( # 2) | 13630.60( # 1) |
| gr431 | 2516.25( # 4) | 2516.25( # 4) | 2214.43( # 2) | 2516.25( # 4) | 2263.78( # 3) | 2153.00( # 1) |
| pr439 | 131282.09( # 3) | 131282.09( # 3) | 130067.88( # 2) | 137778.50( # 6) | 134814.09( # 5) | 118498.55( # 1) |
| pcb442 | 61984.05( # 5) | 61984.05( # 5) | 60891.83( # 3) | 61234.77( # 4) | 58945.83( # 2) | 54291.55( # 1) |
| d493 | 43646.38( # 4) | 43646.38( # 4) | 39982.31( # 2) | 43710.70( # 6) | 43050.15( # 3) | 38869.88( # 1) |
| att532 | 112099.45( # 4) | 112099.45( # 4) | 102201.61( # 2) | 112099.45( # 4) | 103710.35( # 3) | 99101.62( # 1) |
| ali535 | 2671.07( # 4) | 2671.07( # 4) | 2366.50( # 2) | 2671.07( # 4) | 2483.48( # 3) | 2269.30( # 1) |
| u574 | 46881.87( # 4) | 46881.87( # 4) | 44144.83( # 2) | 46881.87( # 4) | 46620.28( # 3) | 39154.50( # 1) |
| rat575 | 8449.32( # 5) | 8449.32( # 5) | 7853.86( # 3) | 8430.71( # 4) | 7808.10( # 2) | 7398.48( # 1) |
| p654 | 43411.56( # 4) | 43411.56( # 4) | 40418.68( # 2) | 48824.31( # 6) | 41009.57( # 3) | 38249.53( # 1) |
| d657 | 62176.40( # 3) | 62176.40( # 3) | 57906.66( # 2) | 62176.40( # 3) | 63518.97( # 6) | 54801.59( # 1) |
| gr666 | 4110.90( # 4) | 4110.90( # 4) | 3670.13( # 2) | 4110.90( # 4) | 3920.88( # 3) | 3507.73( # 1) |
| u724 | 55223.20( # 5) | 55223.20( # 5) | 50245.77( # 3) | 52482.39( # 4) | 49538.41( # 2) | 47108.98( # 1) |
| rat783 | 11255.07( # 4) | 11255.07( # 4) | 10301.88( # 2) | 11255.07( # 4) | 11151.44( # 3) | 9468.73( # 1) |
| dsj1000 | 24630960.10( # 4) | 24630960.10( # 4) | 22291166.04( # 2) | 24630960.10( # 4) | 23977443.37( # 3) | 21147745.60( # 1) |
| pr1002 | 315596.59( # 3) | 315596.59( # 3) | 302938.90( # 2) | 325311.07( # 6) | 320713.45( # 5) | 277925.10( # 1) |
| u1060 | 281635.68( # 5) | 281635.68( # 5) | 270377.38( # 2) | 280689.98( # 4) | 270485.28( # 3) | 241651.40( # 1) |
| vm1084 | 301469.23( # 4) | 301469.23( # 4) | 277435.70( # 3) | 302111.10( # 6) | 273437.36( # 2) | 263501.02( # 1) |
| pcb1173 | 70277.94( # 3) | 70277.94( # 3) | 69010.68( # 2) | 70279.94( # 5) | 71441.00( # 6) | 62453.05( # 1) |
| d1291 | 59941.24( # 3) | 59941.24( # 3) | 59956.75( # 5) | 59892.04( # 2) | 62211.60( # 6) | 55389.49( # 1) |
| rl1304 | 339797.47( # 5) | 339797.47( # 5) | 314295.61( # 3) | 335160.35( # 4) | 303550.70( # 2) | 284242.81( # 1) |
| rl1323 | 332094.97( # 3) | 332094.97( # 3) | 341512.46( # 6) | 331159.20( # 2) | 334237.62( # 5) | 299654.18( # 1) |
| nrw1379 | 70015.46( # 5) | 70015.46( # 5) | 66216.21( # 2) | 69794.61( # 4) | 68362.91( # 3) | 61838.87( # 1) |
| fl1400 | 26971.88( # 4) | 26971.88( # 4) | 22955.32( # 2) | 27057.04( # 6) | 23757.04( # 3) | 22420.49( # 1) |
| u1432 | 188815.01( # 4) | 188815.01( # 4) | 171110.62( # 2) | 196453.25( # 6) | 173995.52( # 3) | 167338.88( # 1) |

Table 7: Algorithm execution time and rank (#) across different TSBLIB instances. Unit: s.

|  | Greedy | NN | Insertion | EoH-TSP | MEoH-TSP | AutoMOAE-TSP |
|---|---|---|---|---|---|---|
| burma14 | 0.00( # 2) | 0.00( # 1) | 0.00( # 3) | 0.05( # 5) | 0.00( # 4) | 0.06( # 6) |
| ulysses16 | 0.00( # 2) | 0.00( # 1) | 0.00( # 3) | 0.02( # 5) | 0.00( # 4) | 0.06( # 6) |
| ulysses22 | 0.00( # 2) | 0.00( # 1) | 0.00( # 4) | 0.04( # 5) | 0.00( # 3) | 0.07( # 6) |
| att48 | 0.00( # 1) | 0.00( # 2) | 0.02( # 4) | 0.07( # 5) | 0.00( # 3) | 0.14( # 6) |
| eil51 | 0.00( # 1) | 0.00( # 2) | 0.03( # 4) | 0.07( # 5) | 0.01( # 3) | 0.16( # 6) |
| berlin52 | 0.00( # 2) | 0.00( # 1) | 0.03( # 4) | 0.07( # 5) | 0.00( # 3) | 0.17( # 6) |
| st70 | 0.00( # 1) | 0.00( # 2) | 0.07( # 4) | 0.09( # 5) | 0.01( # 3) | 0.16( # 6) |
| pr76 | 0.00( # 2) | 0.00( # 1) | 0.07( # 4) | 0.08( # 5) | 0.01( # 3) | 0.18( # 6) |
| eil76 | 0.00( # 2) | 0.00( # 1) | 0.07( # 4) | 0.08( # 5) | 0.01( # 3) | 0.17( # 6) |
| gr96 | 0.00( # 2) | 0.00( # 1) | 0.14( # 5) | 0.12( # 4) | 0.01( # 3) | 0.32( # 6) |
| rat99 | 0.00( # 2) | 0.00( # 1) | 0.15( # 5) | 0.12( # 4) | 0.01( # 3) | 0.29( # 6) |
| kroC100 | 0.00( # 2) | 0.00( # 1) | 0.16( # 5) | 0.12( # 4) | 0.01( # 3) | 0.24( # 6) |
| kroE100 | 0.00( # 2) | 0.00( # 1) | 0.15( # 5) | 0.12( # 4) | 0.01( # 3) | 0.26( # 6) |
| rd100 | 0.00( # 1) | 0.00( # 2) | 0.15( # 5) | 0.12( # 4) | 0.01( # 3) | 0.28( # 6) |
| kroA100 | 0.00( # 2) | 0.00( # 1) | 0.16( # 5) | 0.12( # 4) | 0.01( # 3) | 0.29( # 6) |
| kroB100 | 0.00( # 1) | 0.00( # 2) | 0.16( # 5) | 0.12( # 4) | 0.01( # 3) | 0.27( # 6) |
| kroD100 | 0.00( # 2) | 0.00( # 1) | 0.16( # 5) | 0.12( # 4) | 0.01( # 3) | 0.26( # 6) |
| eil101 | 0.00( # 1) | 0.00( # 2) | 0.16( # 5) | 0.12( # 4) | 0.01( # 3) | 0.30( # 6) |
| lin105 | 0.00( # 2) | 0.00( # 1) | 0.19( # 5) | 0.12( # 4) | 0.01( # 3) | 0.36( # 6) |
| pr107 | 0.00( # 2) | 0.00( # 1) | 0.19( # 5) | 0.13( # 4) | 0.01( # 3) | 0.21( # 6) |
| pr124 | 0.00( # 2) | 0.00( # 1) | 0.31( # 6) | 0.15( # 4) | 0.01( # 3) | 0.29( # 5) |
| bier127 | 0.00( # 1) | 0.00( # 2) | 0.33( # 5) | 0.16( # 4) | 0.01( # 3) | 0.39( # 6) |
| ch130 | 0.00( # 2) | 0.00( # 1) | 0.35( # 6) | 0.17( # 4) | 0.01( # 3) | 0.32( # 5) |
| pr136 | 0.00( # 1) | 0.00( # 2) | 0.41( # 6) | 0.18( # 4) | 0.01( # 3) | 0.35( # 5) |
| gr137 | 0.00( # 2) | 0.00( # 1) | 0.41( # 5) | 0.18( # 4) | 0.01( # 3) | 0.75( # 6) |
| pr144 | 0.00( # 2) | 0.00( # 1) | 0.48( # 6) | 0.19( # 4) | 0.01( # 3) | 0.28( # 5) |
| kroB150 | 0.00( # 2) | 0.00( # 1) | 0.54( # 5) | 0.20( # 4) | 0.01( # 3) | 0.65( # 6) |
| ch150 | 0.00( # 1) | 0.00( # 2) | 0.55( # 5) | 0.21( # 4) | 0.01( # 3) | 0.58( # 6) |
| kroA150 | 0.00( # 2) | 0.00( # 1) | 0.53( # 5) | 0.20( # 4) | 0.01( # 3) | 0.68( # 6) |
| pr152 | 0.00( # 1) | 0.00( # 2) | 0.55( # 6) | 0.21( # 4) | 0.01( # 3) | 0.44( # 5) |
| u159 | 0.00( # 2) | 0.00( # 1) | 0.65( # 6) | 0.23( # 4) | 0.01( # 3) | 0.57( # 5) |
| rat195 | 0.01( # 2) | 0.01( # 1) | 1.18( # 6) | 0.31( # 4) | 0.01( # 3) | 0.88( # 5) |
| d198 | 0.01( # 1) | 0.01( # 2) | 1.23( # 6) | 0.31( # 4) | 0.01( # 3) | 0.96( # 5) |
| kroA200 | 0.01( # 2) | 0.01( # 1) | 1.27( # 5) | 0.31( # 4) | 0.02( # 3) | 1.31( # 6) |
| kroB200 | 0.01( # 2) | 0.01( # 1) | 1.25( # 6) | 0.30( # 4) | 0.01( # 3) | 1.13( # 5) |
| gr202 | 0.01( # 2) | 0.01( # 1) | 1.29( # 5) | 0.31( # 4) | 0.01( # 3) | 1.48( # 6) |
| ts225 | 0.01( # 2) | 0.01( # 1) | 1.76( # 6) | 0.37( # 4) | 0.02( # 3) | 1.06( # 5) |
| tsp225 | 0.01( # 2) | 0.01( # 1) | 1.77( # 5) | 0.36( # 4) | 0.02( # 3) | 1.97( # 6) |
| pr226 | 0.01( # 2) | 0.01( # 1) | 1.80( # 6) | 0.37( # 4) | 0.02( # 3) | 0.93( # 5) |
| gr229 | 0.01( # 1) | 0.01( # 2) | 1.87( # 6) | 0.37( # 4) | 0.02( # 3) | 1.66( # 5) |
| gil262 | 0.01( # 2) | 0.01( # 1) | 2.78( # 6) | 0.46( # 4) | 0.02( # 3) | 2.60( # 5) |
| pr264 | 0.01( # 2) | 0.01( # 1) | 2.83( # 6) | 0.47( # 4) | 0.02( # 3) | 1.22( # 5) |
| a280 | 0.01( # 1) | 0.01( # 2) | 3.40( # 6) | 0.51( # 4) | 0.02( # 3) | 2.01( # 5) |
| pr299 | 0.01( # 1) | 0.01( # 2) | 4.15( # 6) | 0.58( # 4) | 0.02( # 3) | 2.46( # 5) |
| lin318 | 0.02( # 2) | 0.02( # 1) | 4.97( # 6) | 0.64( # 4) | 0.02( # 3) | 3.39( # 5) |
| linhp318 | 0.02( # 1) | 0.02( # 2) | 5.08( # 6) | 0.63( # 4) | 0.02( # 3) | 3.46( # 5) |
| rd400 | 0.02( # 1) | 0.03( # 2) | 9.91( # 6) | 0.92( # 4) | 0.03( # 3) | 6.82( # 5) |
| fl417 | 0.03( # 2) | 0.03( # 1) | 11.60( # 5) | 0.99( # 4) | 0.03( # 3) | 12.69( # 6) |
| gr431 | 0.03( # 2) | 0.03( # 1) | 12.77( # 6) | 1.05( # 4) | 0.03( # 3) | 8.38( # 5) |
| pr439 | 0.03( # 2) | 0.03( # 1) | 13.42( # 6) | 1.08( # 4) | 0.03( # 3) | 7.19( # 5) |
| pcb442 | 0.03( # 2) | 0.03( # 1) | 13.74( # 6) | 1.08( # 4) | 0.03( # 3) | 7.25( # 5) |
| d493 | 0.04( # 2) | 0.04( # 1) | 18.66( # 6) | 1.31( # 4) | 0.05( # 3) | 12.76( # 5) |
| att532 | 0.04( # 1) | 0.04( # 2) | 23.36( # 6) | 1.50( # 4) | 0.05( # 3) | 18.22( # 5) |
| ali535 | 0.04( # 2) | 0.04( # 1) | 24.66( # 6) | 1.52( # 4) | 0.05( # 3) | 21.88( # 5) |
| u574 | 0.05( # 3) | 0.05( # 1) | 29.48( # 6) | 1.71( # 4) | 0.05( # 2) | 24.23( # 5) |
| rat575 | 0.05( # 2) | 0.05( # 1) | 30.47( # 6) | 1.72( # 4) | 0.05( # 3) | 17.80( # 5) |
| p654 | 0.06( # 2) | 0.06( # 1) | 44.83( # 6) | 2.14( # 4) | 0.06( # 1) | 33.29( # 5) |
| d657 | 0.06( # 2) | 0.06( # 3) | 43.99( # 6) | 2.17( # 4) | 0.06( # 1) | 27.59( # 5) |
| gr666 | 0.07( # 3) | 0.06( # 2) | 46.96( # 5) | 2.24( # 4) | 0.06( # 1) | 47.61( # 6) |
| u724 | 0.08( # 3) | 0.08( # 2) | 60.90( # 6) | 2.56( # 4) | 0.07( # 1) | 34.95( # 5) |
| rat783 | 0.09( # 3) | 0.09( # 2) | 75.48( # 6) | 3.01( # 4) | 0.07( # 1) | 67.56( # 5) |
| dsj1000 | 0.15( # 2) | 0.15( # 3) | 162.24( # 6) | 4.67( # 4) | 0.11( # 1) | 160.58( # 5) |
| pr1002 | 0.15( # 2) | 0.15( # 3) | 163.05( # 6) | 4.70( # 4) | 0.11( # 1) | 102.30( # 5) |
| u1060 | 0.16( # 2) | 0.17( # 3) | 191.92( # 6) | 5.20( # 4) | 0.12( # 1) | 138.68( # 5) |
| vm1084 | 0.17( # 2) | 0.18( # 3) | 211.03( # 6) | 5.42( # 4) | 0.12( # 1) | 104.16( # 5) |
| pcb1173 | 0.21( # 2) | 0.21( # 2) | 263.46( # 6) | 6.28( # 4) | 0.14( # 1) | 136.74( # 5) |
| d1291 | 0.25( # 3) | 0.25( # 2) | 354.15( # 6) | 7.47( # 4) | 0.16( # 1) | 52.08( # 5) |
| rl1304 | 0.26( # 2) | 0.26( # 3) | 368.57( # 6) | 7.66( # 4) | 0.16( # 1) | 153.49( # 5) |
| rl1323 | 0.27( # 3) | 0.27( # 2) | 382.37( # 6) | 7.86( # 4) | 0.16( # 1) | 130.90( # 5) |
| nrw1379 | 0.29( # 3) | 0.29( # 2) | 433.28( # 6) | 8.45( # 4) | 0.17( # 1) | 272.26( # 5) |
| fl1400 | 0.30( # 3) | 0.30( # 2) | 452.88( # 5) | 8.66( # 4) | 0.17( # 1) | 538.49( # 6) |
| u1432 | 0.31( # 2) | 0.31( # 3) | 481.20( # 6) | 9.04( # 4) | 0.19( # 1) | 234.80( # 5) |

A.5   DETAILS OF ALGORITHMS FOR GCP AND TSP EVOLVED BY AUTOMOEA AND BASELINES

This section presents the code for the algorithms developed by AutoMOEA and the various baseline methods for the Graph Coloring Problem (GCP) and the Traveling Salesperson Problem (TSP).

**The AutoMOEA-evolved algorithm for the GCP.**

```python
def graph_coloring_v5(adj_matrix):

    import heapq
    n = adj_matrix.shape[0]
    colors = np.full(n, -1)
    degrees = np.sum(adj_matrix, axis=1)
    adj_lists = [np.where(adj_matrix[i] == 1)[0] for i in range(n)]
    adj_colors = [set() for _ in range(n)]
    vertex_heap = [(0, -degrees[i], i) for i in range(n)]
    heapq.heapify(vertex_heap)
    colored = np.zeros(n, dtype=bool)

    def update_saturation(vertex):
        saturation = len(adj_colors[vertex])
        return (-saturation, -degrees[vertex], vertex)

    def update_neighbors(vertex, color):
        for neighbor in adj_lists[vertex]:
            if not colored[neighbor]:
                adj_colors[neighbor].add(color)

    while vertex_heap:
        _, _, vertex = heapq.heappop(vertex_heap)
        if colored[vertex]:
            continue
        used_colors = adj_colors[vertex]
        available_colors = set(range(n)) - used_colors

        if available_colors:
            color_usage = np.zeros(n, dtype=int)
            for neighbor in adj_lists[vertex]:
                if colored[neighbor]:
                    color_usage[colors[neighbor]] += 1
            min_color = min(available_colors, key=lambda c: (color_usage[c], c))
            colors[vertex] = min_color
            colored[vertex] = True

            update_neighbors(vertex, min_color)
            for neighbor in adj_lists[vertex]:
                if not colored[neighbor]:
                    heapq.heappush(vertex_heap, update_saturation(neighbor))

    return colors
```

Figure 5: The AutoMOEA-evolved algorithm for the GCP.

## The algorithm developed by EoH for the GCP.

```python
def graph_coloring_v2(adj_matrix):

    n = adj_matrix.shape[0]
    colors = np.full(n, -1)

    def get_saturation_degree(vertex):
        return len(set(colors[v] for v in range(n) if adj_matrix[vertex][v] == 1 and
colors[v] != -1))

    def find_next_vertex():
        max_saturation = -1
        candidate_vertex = -1
        for vertex in range(n):
            if colors[vertex] == -1:
                saturation_degree = get_saturation_degree(vertex)
                degree = sum(adj_matrix[vertex])
                if  saturation_degree  >  max_saturation  or  (saturation_degree  ==
max_saturation and degree > sum(adj_matrix[candidate_vertex]) if candidate_vertex !=
-1 else degree):
                    max_saturation = saturation_degree
                    candidate_vertex = vertex
        return candidate_vertex

    for _ in range(n):
        u = find_next_vertex()
        if u == -1:
            break

        available_colors = set(range(n))
        for v in range(n):
            if adj_matrix[u][v] == 1 and colors[v] != -1:
                available_colors.discard(colors[v])

        if available_colors:
            color_usage = np.zeros(n, dtype=int)
            for v in range(n):
                if adj_matrix[u][v] == 1 and colors[v] != -1:
                    color_usage[colors[v]] += 1

            min_color = min(available_colors, key=lambda c: (color_usage[c], c))
            colors[u] = min_color

    return colors
```

Figure 6: The EoH-evolved algorithm for the GCP.

## The algorithm developed by MEoH for the GCP.

```python
def graph_coloring_meoh(adjacency_matrix):
    """
    A simple greedy algorithm for graph coloring.

    Args:
        adjacency_matrix: A numpy array representing the adjacency matrix of the
    graph.

    Returns:
        A list of integers where the i-th element represents the color of the i-th vertex.
    """
    n = len(adjacency_matrix)
    colors = [-1] * n

    def is_safe(vertex, color):
        for neighbor in np.where(adjacency_matrix[vertex] == 1)[0]:
            if colors[neighbor] == color:
                return False
        return True

    def backtrack(vertex):
        if vertex == n:
            return True
        for color in range(n):
            if is_safe(vertex, color):
                colors[vertex] = color
                if backtrack(vertex + 1):
                    return True
                colors[vertex] = -1
        return False

    backtrack(0)
    return colors
```

Figure 7: The MEoH-evolved algorithm for the GCP.

**The algorithm developed by Funsearch for the GCP.**

```python
def graph_coloring_v1(adj_matrix):

    n = adj_matrix.shape[0]
    colors = [-1] * n
    degree = np.sum(adj_matrix, axis=1)
    nodes = np.argsort(-degree)

    for node in nodes:
        available_colors = [True] * n

        for neighbor in range(n):
            if adj_matrix[node][neighbor] == 1 and colors[neighbor] != -1:
                available_colors[colors[neighbor]] = False

        for color in range(n):
            if available_colors[color]:
                colors[node] = color
                break

    return colors
```

Figure 8: The Funsearch-evolved algorithm for the GCP.

## The algorithm developed by AutoMOEA for the TSP.

```python
def tsp_04(distances):

    import random
    import math
    n = len(distances)

    unvisited = set(range(1, n))
    current = 0
    route = [0]
    while unvisited:
        next_city = min(unvisited, key=lambda x: distances[current][x])
        route.append(next_city)
        unvisited.remove(next_city)
        current = next_city

    temp = 100.0
    cooling = 0.95
    iterations = 30

    current_cost = calculate_cost(route, distances)
    best_route = route[:]
    best_cost = current_cost

    while temp > 0.1:
        for _ in range(iterations):
            i, j = random.sample(range(1, n), 2)
            new_route = route[:]
            new_route[i], new_route[j] = new_route[j], new_route[i]

            new_cost = calculate_cost(new_route, distances)

            if new_cost < current_cost or random.random() < math.exp((current_cost - new_cost) / temp):
                route = new_route
                current_cost = new_cost
                if new_cost < best_cost:
                    best_route = new_route[:]
                    best_cost = new_cost

        temp *= cooling

    best_route = two_opt(best_route, distances)

    return best_route
```

Figure 9: The AutoMOEA-evolved algorithm for the TSP.

**The algorithm developed by EoH for the TSP.**

```python
def tsp_02(distances: np.ndarray) -> List[int]:

    from typing import List
    import numpy as np
    import random

    n = len(distances)
    path = list(range(n))
    for i in range(n-1):
        min_j = min(range(i+1, n),
                key=lambda j: distances[path[i]][path[j]])
        path[i+1], path[min_j] = path[min_j], path[i+1]

    pop_size = 20
    generations = 100
    population = [path[:]]

    for _ in range(pop_size - 1):
        new_path = path[:]
        i, j = random.sample(range(1, n), 2)
        new_path[i], new_path[j] = new_path[j], new_path[i]
        population.append(new_path)

    for _ in range(generations):
        population.sort(key=lambda x: sum(distances[x[i]][x[i+1]]
                            for i in range(n-1)) + distances[x[-1]][x[0]])
        population = population[:pop_size//2]

        while len(population) < pop_size:
            p1, p2 = random.sample(population, 2)
            cut = random.randint(1, n-2)
            child = p1[:cut]
            child.extend(x for x in p2 if x not in child)
            population.append(child)

    return min(population,
            key=lambda x: sum(distances[x[i]][x[i+1]]
                        for i in range(n-1)) + distances[x[-1]][x[0]])
```

Figure 10: The EoH-evolved algorithm for the TSP.

**The algorithm developed by MEoH for the TSP.**

```python
def tsp_06(distances: np.ndarray) -> List[int]:
    import numpy as np

    def generate_neighborhood_matrix(distance_matrix):
        n = len(distance_matrix)
        neighborhood_matrix = np.zeros((n, n), dtype=int)
        for i in range(n):
            sorted_indices = np.argsort(distance_matrix[i])
            neighborhood_matrix[i] = sorted_indices
        return neighborhood_matrix

    def select_next_node(current_node: int, destination_node: int, unvisited_nodes:
    np.ndarray, distance_matrix: np.ndarray) -> int:
        current_dist = distance_matrix[current_node, unvisited_nodes]
        dest_dist = distance_matrix[destination_node, unvisited_nodes]

        # Normalize distances
        norm_current = current_dist / np.max(current_dist)
        norm_dest = dest_dist / np.max(dest_dist)

        # Weighted score (higher weight for proximity to current node)
        score = 0.7 * norm_current + 0.3 * (1 - norm_dest)

        return unvisited_nodes[np.argmin(score)]

    n = len(distances)
    neighbor_matrix = generate_neighborhood_matrix(distances)
    route = np.zeros(n, dtype=int)
    current_node = 0
    destination_node = 0
    for i in range(1, n - 1):
        near_nodes = neighbor_matrix[current_node][1:]
        mask = ~np.isin(near_nodes, route[:i])
        unvisited_near_nodes = near_nodes[mask]
        next_node     =     select_next_node(current_node,     destination_node,
        unvisited_near_nodes, distances)
        current_node = next_node
        route[i] = current_node
    mask = ~np.isin(np.arange(n), route[:n - 1])
    last_node = np.arange(n)[mask]
    route[n - 1] = last_node[0]
    return route.tolist()
```

Figure 11: The MEoH-evolved algorithm for the TSP.

