# OpenReview forum: "AutoMOAE: Multi-Objective Auto-Algorithm Evolution"
_ICLR.cc/2026/Conference — ICLR 2026 Conference Withdrawn Submission_

### Official Review · Reviewer_MdBq · 2025-10-20

**Soundness:** 2
**Presentation:** 3
**Contribution:** 2
**Rating:** 4
**Confidence:** 4

**Summary:**

This paper introduces AutoMOAE (Auto-Algorithm Evolution for Multi-Objective Requirements), a framework that integrates large language models with evolutionary optimization to enable automatic algorithm design that simultaneously enhances both performance and computational efficiency.
Unlike previous LLM-driven algorithm generation frameworks that focus on a single objective, AutoMOAE explicitly incorporates analytical modules within its crossover and mutation operations and employs a Pareto-based population maintenance strategy to achieve a balance between global multi-objective optimization and local directional guidance.
Experiments on the Graph Coloring Problem and the Traveling Salesman Problem  demonstrate that the algorithms produced by AutoMOAE can match or even surpass both expert-designed and existing automated frameworks in terms of solution quality and efficiency. Ablation and diversity analyses further confirm the effectiveness of the analytical modules and the increased structural complexity of the evolved algorithms.

**Strengths:**

1. The paper addresses a common limitation in current LLM-driven algorithm design—the lack of multi-objective balance—and highlights the necessity of jointly optimizing solution quality and computational efficiency in automatic algorithm generation. The motivation is well grounded and relevant to practical applications.
2. The proposed AutoMOAE framework is structurally complete, covering key components such as initialization, crossover/mutation, and Pareto-front maintenance, with a reasonably clear overall logic.
3. Although the framework is largely built upon existing LLM-based evolutionary approaches, the introduction of an analytical module into the crossover and mutation processes provides a structural enhancement that guides multi-objective optimization, rather than merely restating prior designs.
4. The manuscript is generally well written, with coherent organization and accurate articulation of core concepts.

**Weaknesses:**

1. The framework is validated only on two combinatorial optimization problems (GCP and TSP), which is insufficient to demonstrate its general applicability to more complex tasks.
2. The convergence and stability of AutoMOAE are discussed solely from an empirical perspective, lacking a formal analysis of multi-objective evolutionary convergence or theoretical bounds on computational complexity.
3. Experiments rely exclusively on GPT-4o-mini, without demonstrating the framework’s robustness across different LLM capabilities.
4. Algorithmic complexity is measured only by the number of uncommented code lines, which is intuitive but not rigorous. More informative metrics should be introduced.

**Questions:**

1. Can AutoMOAE be transferred to other algorithm design domains? If so, would the prompting templates need to be redesigned for each domain?
2. The paper states that the “analytical module” guides multi-objective optimization during crossover and mutation. Could the authors elaborate on how dominant objectives are selected when conflicts occur? Is there any heuristic or dynamic weighting mechanism involved?
3. Given the rapid advancement of LLMs (e.g., the imminent release of GPT-6), would AutoMOAE remain applicable if stronger reasoning models were substituted? Relying solely on GPT-4o-mini limits the persuasiveness of the results.
4. Could the authors provide more detailed explanations of the algorithm diversity evaluation? For instance, were the keywords extracted by an LLM? Were mutual information or semantic distance measures used to quantify diversity?
5. In Appendix A.4, the phrase “Figures 6 and 7” appears to be incorrect and should be replaced with “Tables 6 and 7.”

---

### Official Review · Reviewer_R7pc · 2025-10-30

**Soundness:** 2
**Presentation:** 2
**Contribution:** 2
**Rating:** 2
**Confidence:** 4

**Summary:**

This paper proposes an auto-algorithm evolution through LLM for mono-objective problems by considering the Pareto dominance criterion between different requirements.

**Strengths:**

Considering multiple objectives for mono-objective problems sounds interesting.

**Weaknesses:**

It is not very clear to me that what the objective runtime refers to in the Pareto framework. Does it mean how quick the algorithm hits a specific target, or anytime behavior of an algorithm, or the period a algorithm needs under given iterations? If it means the last, I am not quite sure how important this objective is as for evolutionary algorithms involving crossover and mutation, the time needed may not differ very much.

As mentioned by the author(s), there is a similar work MEoH, which used the similar idea, i.e., considering multiple objectives for mono-objective problems through LLM. The novelty of the proposed work only lies in ``embeds analytical modules within its specific crossover and mutation operators''.

Typos: 3.3.1. there should be a comma between "runtime'' and "i.e.''. In the same page, "an offspring" should be "an offspring solution".

**Questions:**

Could you please respond to the first two comments in the Weaknesses field.

---

### Official Review · Reviewer_hMun · 2025-10-31

**Soundness:** 2
**Presentation:** 3
**Contribution:** 3
**Rating:** 4
**Confidence:** 4

**Summary:**

The paper introduces a new automated algorithm design framework, called AutoMOAE, which is designed to address multi-objective optimization problems.
The paper tests AutoMOAE on instances of the GCP and TSP, which are single-objective problems, but the authors treat them as bi-objective problems, where one objective is the quality of the found solution and the other is efficiency.
In my view, AutoMOAE represents a form of "multi-objectivation."

**Strengths:**

REDEFINE crossover and mutation operators in terms of LLM context. It is an interest idea, while it would be good if see something on crossover rate and mutation rate.

**Weaknesses:**

1) A concern is that the algorithm's performance relies on the coding ability of the LLM. Since LLMs vary in their efficiency and accuracy, this could lead to inconsistent results and affect the overall performance of AutoMOAE.

2) Time consumption could be a significant weakness. The algorithm may require considerable computational resources, which could make it less practical for large-scale or real-time applications.

3) The algorithm is tested on relatively simple problems like GCP and TSP. However, it may face challenges when applied to more complex, high-dimensional, or dynamic problem settings, requiring further adaptation.

**Questions:**

Are there any broader test results available, particularly on open-ended or more complex problems?
E.G., A case study on how the algorithm performs in such challenging scenarios would provide valuable insights into its robustness and scalability.

---

### Official Review · Reviewer_Qxdm · 2025-11-01

**Soundness:** 2
**Presentation:** 1
**Contribution:** 1
**Rating:** 2
**Confidence:** 5

**Summary:**

This paper introduces AutoMOAE, a framework for multi-objective algorithm design using LLMs and evolutionary computation. While the research direction is promising, the paper suffers from fundamental weaknesses in its argumentation, experimental analysis, and particularly in its comparison to MEoH.

**Strengths:**

The paper addresses the important and challenging frontier of automated algorithm design.

**Weaknesses:**

1. Inadequate Introduction and Positioning: The introduction severely lacks a proper overview of related problems and existing work. Critically, it fails to provide the necessary context and positioning for its primary competitor, MEoH, undermining the entire paper's foundational argument.
2. The comparison to MEoH is vague, lacks depth, and fails to establish novelty: This is the paper's most critical flaw.
      ○ Lack of Explanation of MEoH's Mechanism: The paper critiques MEoH’s “strength-diversity mechanism” without ever explaining how it works. This reduces MEoH to a "strawman" and prevents readers from assessing whether AutoMOAE's "local guidance" is a genuine paradigm shift.
      ○ False Dichotomy of "Macro" vs. "Local": This distinction is unconvincing, as AutoMOAE itself employs a "macro" Pareto-based selection strategy. The paper fails to argue why its "local guidance" is fundamentally different from or superior to MEoH's approach.
      ○ Experimental Results Do Not Support the Argument: On the TSP benchmark (Table 1), AutoMOAE is significantly slower than MEoH (runtime rank 5.39 vs. 2.92). This indicates a severe computational overhead from its "local guidance," a critical trade-off the paper fails to analyze.
3. Exaggerated Claims of Contribution and Performance: The abstract claims the method "consistently matches or surpasses" solutions in "both solution quality and efficiency." This is directly contradicted by the TSP runtime results. The paper must state its contributions more accurately and acknowledge the clear efficiency trade-offs.
4. Superficial Measure of Algorithmic Complexity: Using "uncommented lines of code"  as a proxy for complexity is naive and misleading.  More rigorous metrics (e.g., cyclomatic complexity) are required to support this claim.
5. Shallow Assessment of Diversity: The analysis based on keyword counts is superficial. A higher number of keywords does not equate to more effective exploration of the design space. A stronger analysis would link this diversity to improvements on the Pareto front.

**Questions:**

How do you prove your "local guidance" is mechanistically superior to the MEoH mechanism, which you never explain in detail?

How do you reconcile the claim of "surpassing in efficiency" with the near-worst runtime rank in the TSP experiments?

Why use an unreliable metric like "lines of code" (LOC) for algorithmic complexity instead of a more scientific standard?

Beyond a higher keyword count, what direct evidence shows that this "diversity" leads to a better Pareto front?

---

### Official Review · Reviewer_adga · 2025-11-01

**Soundness:** 3
**Presentation:** 3
**Contribution:** 2
**Rating:** 4
**Confidence:** 3

**Summary:**

Over the last couple of years, numerous attempts have been made at automated evolutionary algorithm design using LLMs. This paper is another one in this direction, proposing a method called AutoMOAE. One of the main ideas in this paper is to consider algorithm search as a multi-objective problem where we want to simultaneously minimise runtime and maximise solution quality.

AutoMOAE has been evaluated on TSP and a graph colouring problem, giving good results compared with existing methods, such as MEoH, EoH, and FunSearch.

**Strengths:**

The approach compares favourably with existing similar frameworks, including EoH, MEoH, and FunSearch. In particular, it yields algorithms with lower average running time while finding good quality solutions.

The main algorithm has been described in detail using pseudo-code.

**Weaknesses:**

There are now already quite a few such LLM-based automated algorithm design frameworks. The authors have not sufficiently highlighted the novelty of this particular system.

A central feature of AutoMOAE is to consider search for algorithms as a multi-objective problem. However, this idea was already proposed before, e.g., in the MEoH paper by Yao et al.

Another idea mentioned are the "analysis components". This concept remains vague, since it is not defined and only mentioned on page 8. The benefit of the "analysis component" is considered in the ablation study.

There is no proper statistical evaluation of the results. Tables and figures report average values without indicating the variance. The algorithm is only run for 10 generations.

**Questions:**

Please list your contributions explicitly in the end of the introduction.

Please explain more clearly the novelty of your approach. What are the novel aspects of AutoMOAE relative to e.g., MEoH?

Define clearly what you mean by "analysis component"

Please carry out a proper statistical evaluation of the experimental results. E.g., is there a statistically significant difference in the performance with and without "the analysis component"? In all figures and tables, please include information about variance.

How are the algorithms evauluated? Do you run the algorithms for a fixed number of function evaluations, or for a fixed number of iterations? Do you run the algorithms multiple times and consider the average runtime or the distribution of the runtime? If not, why not?

Please provide more details about the problem instances in the main text (e.g., number of nodes, graph model, etc.)

Minor comments:

There are several typographical and spelling mistakes. E.g., line 174, select, line 254 runtime, multiple spelling mistakes in Figure 4(b).

The terminology used can be misleading. In evolutionary search, a mutation operator refers to a random variation of a search point without regards of the objective value/fitness. However, your "mutation" prompt also evaluates individuals. You may consider a different name for this operator/prompt.

In Section 3.3.1, you should cite the Pareto front sorting from NSGA-II (Deb et al).

Algorithm 1 also requires a set of benchmark problems to evaluate the algorithms. This is not mentioend in the first line of the algorithm. Also, in line 13, it is unclear what "highest scoring response" refers to. Do you mean the algorithm with lowest runtime, or the algorithm with best solution quality, or something else?

---

### Note · Authors · 2025-11-18

**Comment:**

I have read and agree with the venue's withdrawal policy on behalf of myself and my co-authors.

**Withdrawal Confirmation:**

I have read and agree with the venue's withdrawal policy on behalf of myself and my co-authors.